# Machine Learning: Volume and Biomass Estimates of Commercial Trees in the Amazon Forest

**Samuel José Silva Soares da Rocha** [1,*], **Flora Magdaline Benitez Romero** [2],
**Carlos Moreira Miquelino Eleto Torres** [3], **Laércio Antônio Gonçalves Jacovine** [3], **Sabina Cerruto Ribeiro** [4],
**Paulo Henrique Villanova** [3], **Bruno Leão Said Schettini** [3], **Vicente Toledo Machado de Morais Junior** [5],
**Leonardo Pequeno Reis** [6], **Maria Paula Miranda Xavier Rufino** [3], **Indira Bifano Comini** [3],
**Ivaldo da Silva Tavares Júnior** [3] **and Águida Beatriz Traváglia Viana** [3]

[1] Departamento de Ciências Florestais, Universidade Federal de Lavras, Lavras 37200-900, MG, Brazil
[2] Instituto Nacional de Pesquisas da Amazônia—INPA, Manaus 69067-375, AM, Brazil
[3] Departamento de Engenharia Florestal, Universidade Federal de Viçosa, Viçosa 36570-900, MG, Brazil
[4] Centro de Ciências Biológicas e da Natureza, Universidade Federal do Acre (UFAC),
   Campus Universitário BR 364, Km 04, Distrito Industrial, Rio Branco 69920-900, AC, Brazil
[5] Brandt Meio Ambiente LTDA, Alameda do Ingá, 89, Vale do Sereno, Nova Lima 34006-042, MG, Brazil
[6] Instituto de Desenvolvimento Sustentável Mamirauá, Tefé 69553-225, AM, Brazil
**\*** Correspondence: samuel.rocha@ufla.br

**Abstract:** Accurate estimation of the volume and above-ground biomass of exploitable trees by the practice of selective logging is essential for the elaboration of a sustainable management plan. The objective of this study is to develop machine learning models capable of estimating the volume and biomass of commercial trees in the Southwestern Amazon, based on dendrometric, climatic and topographic characteristics. The study was carried out in the municipality of Porto Acre, Acre state, Brazil. The volume and biomass of sample trees were determined using dendrometric, climatic and topographic variables. The Boruta algorithm was applied to select the best set of variables. Support Vector Machines (SVM), Artificial Neural Networks (ANN), Random Forests (RF) and the Generalized Linear Model (GLM) were the machine learning methods evaluated. In general, the evaluated methods showed a satisfactory generalization power. The results showed that the volume and biomass predictions of commercial trees in the Amazon rainforest differed between the techniques ($p < 0.05$). ANNs showed the best performance in predicting the volume and biomass of commercial trees, with the highest $r_{y\hat{y}}$ and the lowest RSME and MAE. Thus, machine learning methods such as SVM, ANN, RF and GLM are shown to be useful and efficient tools for estimating the volume and biomass of commercial trees in the Amazon rainforest. These methods can be useful tools to improve the accuracy of estimates in forest management plans.

**Keywords:** biome allometry; dense rainforest; models; artificial intelligence; biometry

## 1. Introduction

Machine learning is a rapidly growing area of study in artificial intelligence and is useful in forest modeling because of its potential to produce better models than traditional data modeling approaches [1–3], particularly allometric equations [4–15]. The applications of this computational intelligence technique in the forestry sector have gained great relevance [16–18]. Models have already been efficiently tested to estimate tree growth [19], biomass and carbon [20–22], map species richness and composition [23], predict tree diameter and height [24–26], map tropical forest structure [27] and assess forest quality parameters [28].

The application of these models to estimate tree volume and biomass can be promising for obtaining accurate estimates of these tree parameters. Machine learning algorithms are capable of processing vast amounts of data from various sources, such as remote

sensing and ground-based measurements, to generate accurate estimates of tree volume and biomass. This information is critical for forest management, conservation and carbon accounting efforts in the Amazon.

The Brazilian Amazon rainforest is the largest remnant of the forest [29] and has an important role in the carbon cycle, storing around 150–200 Pg of carbon in biomass and living things [30]. The biome is home to more than a third of all the diversity of neotropical plants [31] and 6700–16,000 tree species [32,33].

Sustainable forest management is critical for ensuring the long-term health and productivity of the forest while meeting the needs of local communities and supporting economic development [34–36]. Accurate estimates of tree volume and biomass are essential for planning and implementing sustainable forest management practices [37,38] such as selective logging and forest restoration [5].

Machine learning (ML) methods have been increasingly used for biomass estimation, as they can handle large datasets, complex relationships and nonlinear patterns in ecological systems. Algorithms such as Support Vector Machines (SVM), Artificial Neural Networks (ANN), Random Forests (RF) and the Generalized Linear Model (GLM) are often used in these studies, seeking more accurate volume and biomass estimates [21,39–42]. For example, researchers have used Random Forests to estimate aboveground biomass using airborne spectral indices [43], and also used Neural Networks and support-vector regression to estimate forest biomass using climate data [44].

This superiority in the generation of estimates is associated with a lower number of assumptions about data and processes [45], which allows the generation of better prediction results, in view of the complex relationships of forest dynamics. In the above context, this study aims to develop machine learning models that are capable of estimating the volume and biomass of commercial trees in the Southwestern Amazon, based on dendrometric, climatic and topographic characteristics. The research questions of this study are: (i) Are the machine learning methods evaluated efficient for estimating the volume and biomass of commercial trees? (ii) What is the best method to estimate the volume of commercial trees? and (iii) What is the best method to estimate the biomass of commercial trees?

To achieve the objectives of this study, a systematic approach was followed in this study. Firstly, relevant literature was reviewed to identify the key variables affecting the volume and biomass of commercial trees in the Southwestern Amazon. Secondly, an extensive dataset was collected, which included dendrometric, climatic and topographic variables, as well as volume and biomass measurements of commercial trees. Thirdly, machine learning models were developed and trained using the collected dataset. Fourthly, the performance of different machine learning methods was evaluated and compared in terms of accuracy and precision. Finally, the best-performing machine learning models were selected, and the results were discussed in light of the research questions.

## 2. Materials and Methods

### 2.1. Characterization of the Study Area

The study was carried out at Antimary Farm I and II, located in the Southwestern Amazon, in the municipality of Porto Acre, Acre, Brazil (Figure 1). The area under sustainable management comprises 1253.02 ha. The region's vegetation is classified as "terra firme"—a forest with solid ground—and wetland rainforest [46]. The climate of the region is of the Am type, according to the Köppen classification [47]. The study area presents two types of soil, Red Argisol and Dystrophic Red Yellow Latosol [48]. The topography is predominantly flat, with a slope of around 5%. The altimetry varies between 220 and 300 m above sea level.

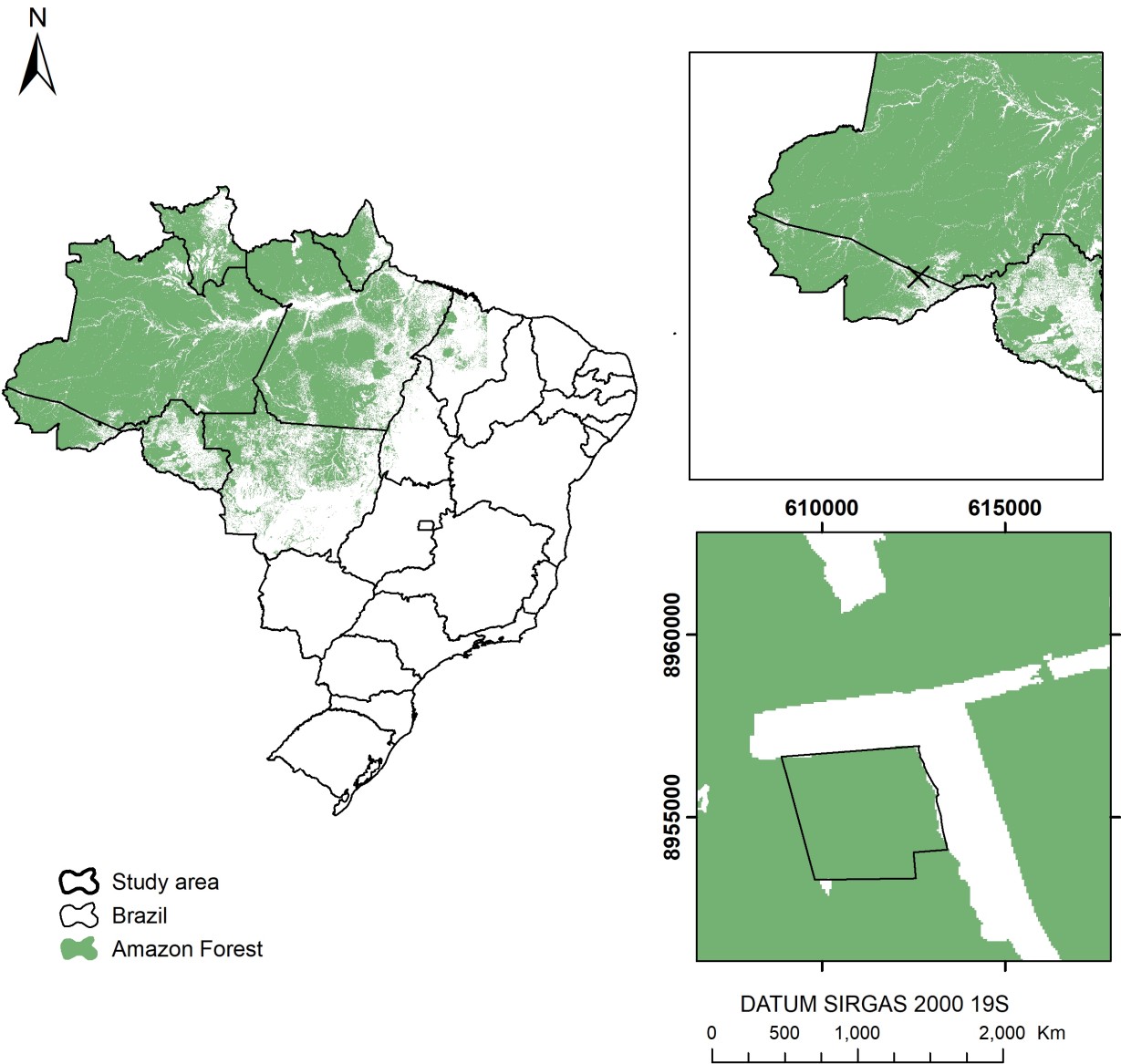

**Figure 1.** Location of the study area in the Southwestern Amazon, in the municipality of Porto Acre, Acre, Brazil.

A census was conducted in the exploitable area in May 2015 and the Sustainable Forest Management Plan (SFMP) was approved in 2016 by the Acre Environment Institute (Instituto de Meio Ambiente do Acre, IMAC).

*2.2. Determination of Volume and Biomass Stocks*

Sample trees were selected based on their density and basal area, obtained from information from the census provided by the company responsible for the management, in which all trees of a commercial interest with a diameter at breast height (DBH; 1.30 m) $\geq$ 50 cm were measured. Eighteen species of the highest importance and value were selected and distributed in 214 individuals [49] (Table 1).

**Table 1.** Number of trees and basic wood density of commercial tree species present in Southwestern Amazon in the municipality of Porto Acre, Acre, Brazil.

| SN | F | N | Bd |
|---|---|---|---|
| *Albizia niopoides* (Spruce ex Benth.) Burkart | *Fabaceae* Lindl. | 7 | 0.64 ± 0.03 |
| *Apuleia leiocarpa* (Vogel) J.F.Macbr. | *Fabaceae* Lindl. | 13 | 0.77 ± 0.03 |
| *Astronium lecointei* Ducke | *Anacardiaceae* R.Br. | 6 | 0.82 ± 0.05 |
| *Barnebydendron riedelii* (Tul.) J.H.Kirkbr. | *Fabaceae* Lindl. | 5 | 0.57 ± 0.03 |
| *Buchenavia tetraphylla* (Aubl.) R.A.Howard | *Combretaceae* R.Br. | 9 | 0.69 ± 0.04 |
| *Castilla ulei* Warb. | *Moraceae* Gaudich. | 37 | 0.41 ± 0.04 |
| *Cedrela odorata* L. | *Meliaceae* A.Juss. | 8 | 0.43 ± 0.04 |
| *Ceiba pentandra* (L.) Gaertn. | *Malvaceae* Juss. | 4 | 0.29 ± 0.03 |
| *Ceiba samauma* (Mart.) K.Schum. | *Malvaceae* Juss. | 22 | 0.51 ± 0.05 |
| *Copaifera multijuga* Hayne | *Fabaceae* Lindl. | 6 | 0.52 ± 0.05 |
| *Dipteryx odorata* (Aubl.) Willd. | *Fabaceae* Lindl. | 11 | 0.80 ± 0.04 |
| *Eschweilera bracteosa* (Poepp. ex O.Berg) Miers | *Lecythidaceae* A.Rich. | 15 | 0.65 ± 0.05 |
| *Eschweilera grandiflora* (Aubl.) Sandwith | *Lecythidaceae* A.Rich. | 13 | 0.73 ± 0.03 |
| *Handroanthus serratifolius* (Vahl) S.Grose | *Bignoniaceae* Juss. | 8 | 0.82 ± 0.04 |
| *Hura crepitans* L. | *Euphorbiaceae* Juss. | 6 | 0.36 ± 0.06 |
| *Hymenaea courbaril* L. | *Fabaceae* Lindl. | 8 | 0.76 ± 0.04 |
| *Parkia paraensis* Ducke | *Fabaceae* Lindl. | 20 | 0.46 ± 0.06 |
| *Schizolobium parahyba* var. *amazonicum* (Huber ex Ducke) Barneby | *Fabaceae* Lindl. | 16 | 0.48 ± 0.08 |
| $\bar{X} \pm$ CI | | 11.89 ± 8.12 | 0.59 ± 0.17 |

Where: SN = Scientific name; F = family; N = number of individuals; Bd = Basic wood density, in g cm$^{-3}$; $\bar{X}$ = Mean; CI = confidence interval.

The volume of the selected individuals was determined by strict cubing using the method of Smalian [50]. Wood disks from the base of the logs were collected to determine the basic wood density according to the ABNT standard (2003) [51]. Biomass was calculated by multiplying the volume and the basic density of the wood [8]. The average basic wood density of the trees analyzed was 0.59 g cm$^{-3}$ (Table 1).

### 2.3. Predictor Variables for Modeling

For the modeling, dendrometric, qualitative, climatic and topographic variables (predictor variables) were used to estimate the volume and biomass stocks (response variables). These variables have been shown to be important drivers of vegetation patterns in many ecosystems including the Amazon. Climatic variables such as temperature and precipitation can affect plant growth and distribution, while topographic variables such as elevation and slope can influence factors such as soil moisture and nutrient availability. The dendrometric variables used were DBH, commercial height (Ch) and basic wood density (Bd) [49]. The qualitative variables were the species and the family of the individuals [49]. The bioclimatic variables used (Bio 1-19) are derived from the monthly values of temperature and precipitation and were obtained from the WorldClim—Global Climate Data database [52], with a spatial resolution of approximately 1 km$^2$. The Bio 5 climatic variable was not used because it did not show variability in the study area. The topographic variable used was altitude (Table 2).

**Table 2.** Predictor variables used in the modeling of volume and biomass in the Southwestern Amazon, in the municipality of Porto Acre, Acre, Brazil.

| Variable | Min | 1° Quartil | Median | Mean | 3° Quartil | Max | SD |
|---|---|---|---|---|---|---|---|
| DBH | 50.38 | 64.78 | 75.44 | 79.60 | 89.52 | 149.92 | 20.14 |
| Ch | 7.30 | 11.71 | 14.20 | 14.82 | 17.87 | 25.40 | 4.03 |
| Bd | 0.29 | 0.43 | 14.20 | 0.57 | 0.73 | 0.82 | 0.15 |
| Bio1 | 24.86 | 24.88 | 24.88 | 24.88 | 24.89 | 24.92 | 0.01 |
| Bio2 | 11.45 | 11.51 | 11.52 | 11.51 | 11.52 | 11.53 | 0.01 |

**Table 2.** *Cont.*

| Variable | Min | 1° Quartil | Median | Mean | 3° Quartil | Max | SD |
|---|---|---|---|---|---|---|---|
| Bio3 | 81.21 | 81.62 | 81.68 | 81.65 | 81.68 | 82.02 | 0.09 |
| Bio4 | 82.28 | 83.38 | 83.42 | 83.49 | 83.63 | 84.39 | 0.25 |
| Bio6 | 17.30 | 17.30 | 17.30 | 17.30 | 17.30 | 17.40 | 0.01 |
| Bio7 | 14.00 | 14.10 | 14.10 | 14.10 | 14.10 | 14.10 | 0.01 |
| Bio8 | 25.32 | 25.32 | 25.33 | 25.33 | 25.35 | 25.38 | 0.02 |
| Bio9 | 23.63 | 23.67 | 23.67 | 23.67 | 23.68 | 23.73 | 0.02 |
| Bio10 | 25.55 | 25.57 | 25.57 | 25.57 | 25.58 | 25.60 | 0.01 |
| Bio11 | 23.63 | 23.67 | 23.67 | 23.67 | 23.68 | 23.73 | 0.02 |
| Bio12 | 1830.00 | 1834.00 | 1836.00 | 1836.31 | 1839.00 | 1853.00 | 3.98 |
| Bio13 | 250.00 | 251.00 | 252.00 | 251.61 | 252.00 | 254.00 | 1.03 |
| Bio14 | 40.00 | 40.00 | 41.00 | 40.70 | 41.00 | 41.00 | 0.46 |
| Bio15 | 51.47 | 51.58 | 51.77 | 51.74 | 51.77 | 52.05 | 0.17 |
| Bio16 | 735.00 | 736.00 | 738.00 | 737.61 | 738.00 | 744.00 | 2.09 |
| Bio17 | 154.00 | 155.00 | 155.00 | 155.29 | 156.00 | 158.00 | 0.91 |
| Bio18 | 568.00 | 570.00 | 570.00 | 570.27 | 571.00 | 576.00 | 1.69 |
| Bio19 | 154.00 | 155.00 | 155.00 | 156.82 | 156.00 | 198.00 | 8.01 |
| Alt | 151.05 | 164.99 | 173.16 | 175.08 | 183.08 | 248.87 | 13.75 |

Where: Min = minimum value; Max = maximum value; SD = standard deviation; DBH = diameter at breast height, in cm; Ch = commercial height, in m; Db = basic wood density; in g cm$^{-3}$; Bio1 = annual mean temperature (°C); Bio2 = mean diurnal range (°C); Bio3 = Isothermality (%); Bio4 = temperature seasonality; Bio5 = max temperature of warmest month (°C); Bio6 = min temperature of coldest month (°C); Bio7 = temperature annual range (°C); Bio8 = mean temperature of wettest quarter (°C); Bio9 = mean temperature of driest quarter (°C); Bio10 = mean temperature of warmest quarter (°C); Bio11 = mean temperature of coldest quarter (°C); Bio12 = annual precipitation (mm); Bio13 = precipitation of wettest month (mm); Bio14 = precipitation of driest month (mm); Bio15 = precipitation seasonality (Coefficient of Variation) (mm); Bio16 = precipitation of wettest quarter (mm); Bio17 = precipitation of driest quarter (mm); Bio18 = precipitation of warmest quarter (mm); Bio19 = precipitation of coldest quarter (mm); Alt = altitude.

The Boruta algorithm [53] was applied to select the best set of predictor variables to estimate volume and biomass. Boruta is a feature selection algorithm that was specifically designed to work with random forest models. The algorithm identifies the most relevant variables in a dataset by comparing the importance of each variable to that of randomly generated shadow variables. The shadow variables are created by permuting the values of each variable so that they are no longer associated with the response variable. The importance of each variable is then assessed by comparing the accuracy of a random forest model trained with the original variables to that of a model trained with the shadow variables. This algorithm iteratively removes the resources that are proven by a statistical test to be less relevant than random probes [53]. The Boruta R Software Package was used.

The quantitative variables were standardized to accelerate the convergence rate and reduce the iteration process in training:

$$Z_i = \left( x_i - \bar{x} \right) / \sigma$$

where:

$Z_i$—standardized value of the i-th observation;

$x_i$—value of the *i*-th observation;

$\bar{x}$—average of the observed values;

$\sigma$—standard deviation;

The scale function of R Software was used in this step.

### 2.4. Model Evaluation

The models tested to estimate the volume and biomass were: Support Vector Machines (SVM), Artificial Neural Networks (ANN), Random Forests (RF) and the Generalized Linear Model (GLM).

The trained ANNs were of the multilayer perceptron (MLP) type. The typical MLP architecture consists of an input layer containing the predictor variables, one or more hidden layers and an output layer containing the predicted variable. The activation function used was logistics. The training algorithms used were resilient propagation. The ANNs were implemented with the MLP function of the "RSNNS" Package in R.

The function svm of the "e1071" Package on R was used for training SVMs. The Kernel function was of the linear type. The random Forest function of the package of the same name in R was used for RF training. The glm function and link function of identity type and Gaussian family were used for GLM.

The performance of the models in the estimation of volume and biomass was assessed using the k–fold cross-validation method, with the data divided into 5 folds (4 for adjustments/training and 1 for validation). The k-fold cross-validation method is a technique used in machine learning to assess the performance of a model. In this method, the data is divided into k equally sized subsets or "folds". The model is then trained on k-1 folds and validated on the remaining fold. This process is repeated k times, with each fold being used as the validation set once. The results are averaged over the k runs to obtain an estimate of the model's performance. At each adjustment/training of the 5 folds, the metrics of Root Mean Square Error—RMSE (Equation (1)) and mean absolute error—MAE (Equation (2)) were calculated. This process was repeated 50 times, obtaining the average of the metrics for comparison of all models. The data were selected randomly in each of the 50 repetitions, resulting in different data sets, for greater robustness of the evaluation.

$$\text{RMSE} = \sqrt{\frac{1}{R}\sum_{r=1}^{R}\frac{\sum_{i=1}^{n}\left(X_i - \hat{X}_i\right)^2}{n}} \tag{1}$$

$$\text{MAE} = \frac{1}{R}\sum_{r=1}^{R}\frac{\left|\sum_{i=1}^{n}\left(X_i - \hat{X}_i\right)\right|}{n} \tag{2}$$

where:

$R$—number of repetitions (50);

$n$—number of observations;

$X_i$—observed variable from the $i$-th tree, in m;

$\hat{X}_i$—estimated variable of the $i$-th tree, in m.

The averages of RMSE and MAE of each method in each repetition were ranked with weight assignments from 1 to 4, with 1 for the lowest value and 4 for the highest value. The weight $p_{ij}$ assigned to the model $m_j$ for the mean of RMSE was added to the weight $p_{ij}$ assigned to the same model $m_j$ for the mean of MAE, with i = 1, 2, . . . , 50. With the result of these sums, the values were submitted to the Friedman–Nemenyi test, at the 5% significance level (Equation (3)).

$$CD = q_\alpha \sqrt{\frac{k(k+1)}{6N}} \tag{3}$$

where:

$CD$—critical difference;

$q_\alpha$—critical value calculated based on Studentized interval statistics (Harter, 1960) divided by $\sqrt{2}$;

$k$—number of algorithms being compared;

$N$—number of data sets.

## 3. Results

### 3.1. Selection of Variables

The applied variable selection procedure allowed the choice of the best model based on the ideal subset of variables. Figure 2a shows the variables that are most important for modeling tree volume, with the size of the bar representing the importance of the variable

and the green color of the barplots indicating which variables have been selected. The figure demonstrates that the diameter at breast height (DBH) and commercial height (Ch) are significant predictors of tree volume. This relationship is expected, as DBH and Ch are common metrics used to estimate tree volume (Figure 2). On the other hand, Figure 2b displays the addition of the wood basic density variable for biomass prediction. The authors note that this finding is consistent with previous research, which has shown that denser trees tend to have higher biomass. The bioclimatic variables and the topographic variable altitude were not considered significant for the modeling of volume and biomass based on the Boruta algorithm (Figure 2).

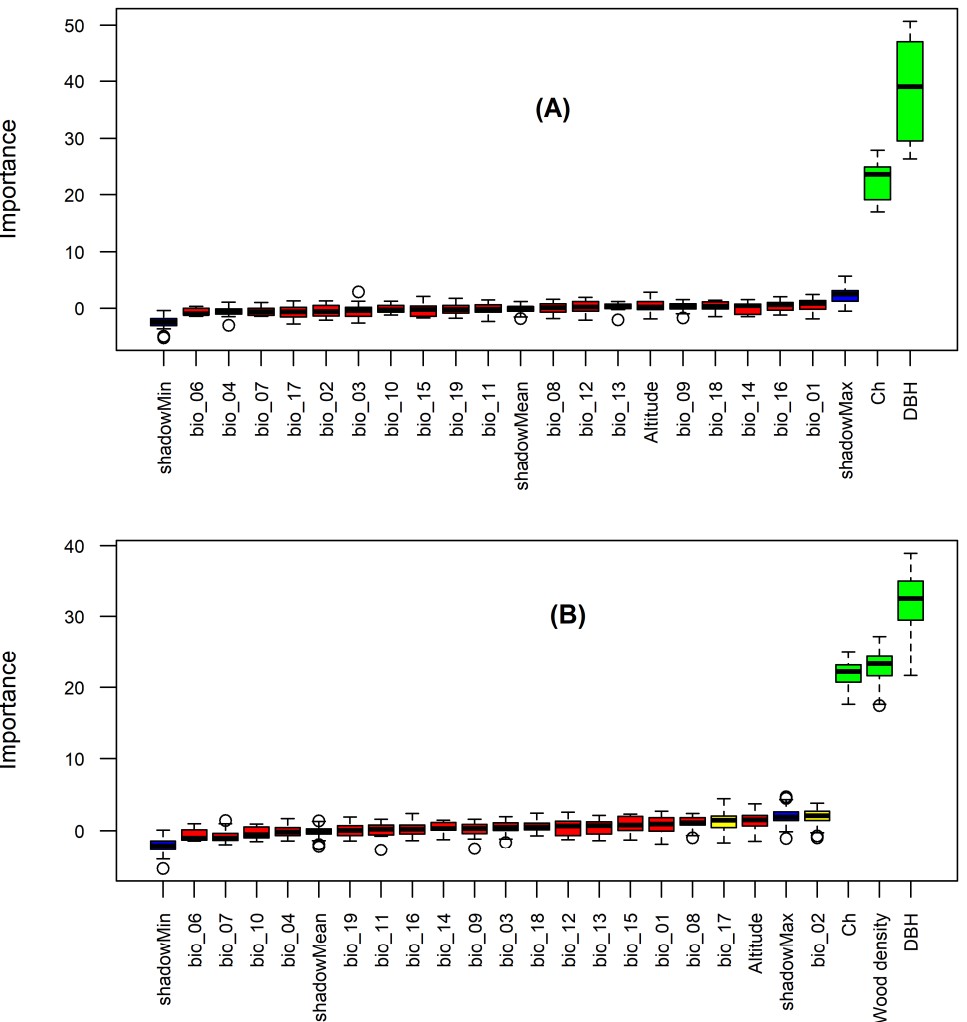

**Figure 2.** Most important variables for modeling the volume (**A**) and biomass (**B**) of commercial trees in Southwestern Amazon, in the municipality of Porto Acre, Acre, Brazil.

### 3.2. Selection of Models

Figure 3 shows the relationship between the observed and predicted values of volume and biomass for each evaluated technique. The blue line represents the tendency of the data. In general, the evaluated models showed a satisfactory generalization power, indicated by similar precision results between the observed and estimated data in the validation for all variables studied (Figure 3). Pearson's correlation coefficient ($r_{y\hat{y}}$) between the estimated and observed volume and biomass data was greater than 0.85 in all the machine learning models applied.

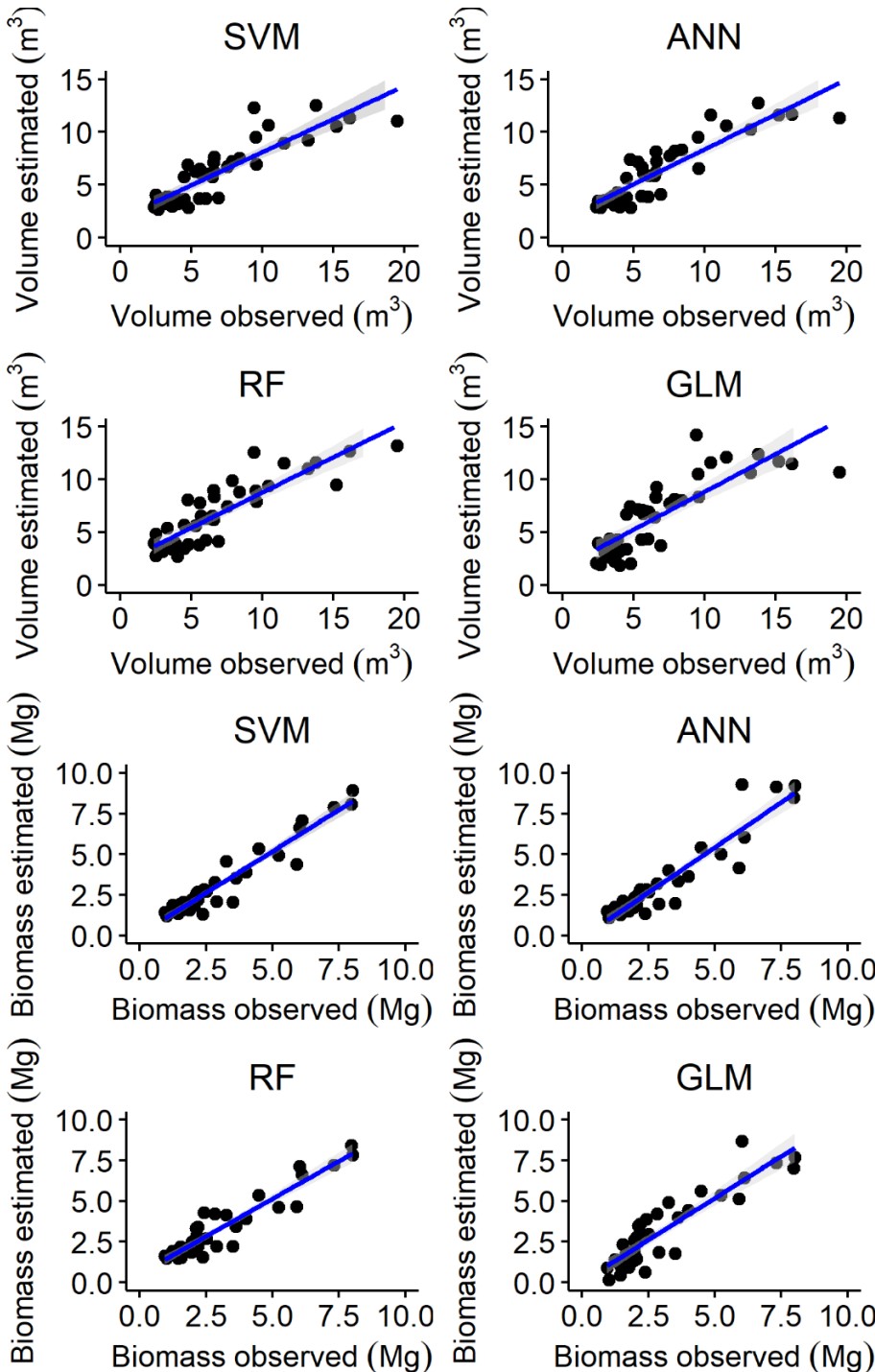

**Figure 3.** Observed and estimated values of volume and biomass by the different machine learning models, SVM, ANN, RF and GLM tested.

GLM showed the best performance to estimate the volume of commercial trees, with the highest $r_{y\hat{y}}$ and the lowest RSME and MAE for all repetitions (Table 3). RF had a $r_{y\hat{y}}$ close to the GLM model and the second-best performance for predicting volume. The ANN algorithm showed moderate performance, and the SVM had the worst performance for predicting the volume of commercial trees in the Amazon.

**Table 3.** Statistics of the tested machine learning models (SVM, ANN, RF and GLM) for modeling volume and biomass of commercial trees in the Southwestern Amazon in the municipality of Porto Acre, Acre, Brazil.

| Variable | Model | RMSE | MAE | $r_{y\hat{y}}$ |
|---|---|---|---|---|
| Volume | SVM | 1.93 ± 0.54 | 1.19 ± 0.23 | 0.89 ± 0.04 |
|  | ANN | 1.67 ± 0.36 | 1.13 ± 0.19 | 0.91 ± 0.04 |
|  | RF | 1.82 ± 0.41 | 1.24 ± 0.20 | 0.90 ± 0.04 |
|  | GLM | 1.82 ± 0.33 | 1.30 ± 0.18 | 0.89± 0.04 |
| Biomass | SVM | 1.15 ± 0.33 | 0.67 ± 0.15 | 0.92 ± 0.03 |
|  | ANN | 1.10 ± 0.27 | 0.69 ± 0.13 | 0.92 ± 0.03 |
|  | RF | 1.19 ± 0.31 | 0.76 ± 0.14 | 0.91 ± 0.03 |
|  | GLM | 1.35 ± 0.27 | 0.97 ± 0.12 | 0.88 ± 0.04 |

Where: RMSE: Root Mean Square Error; MAE: mean absolute error; SVM: Support Vector Machine; ANN: Artificial Neural Networks; RF: Random Forest; GLM: Generalized Linear Model.

ANN showed the best performance for predicting biomass, with the highest $r_{y\hat{y}}$ and the lowest RMSE and MAE for all repetitions. RF also had the second-best performance for predicting biomass. SVM had the worst performance for the prediction of commercial tree biomass in the Southwestern Amazon (Table 3).

The means of RMSE and MAE varied over the repetitions for each technique. The RMSE and MAE averages of ANN and GLM showed the lowest values when estimating the volume of trees (Figure 3). ANN and RF showed the lowest RMSE and MAE over 50 repetitions in the cross-validation when estimating the biomass (Figure 4). SVM showed greater instability in the values of RMSE and MAE of the cross-validation and the highest values of RMSE and MAE for all variables evaluated in the present study.

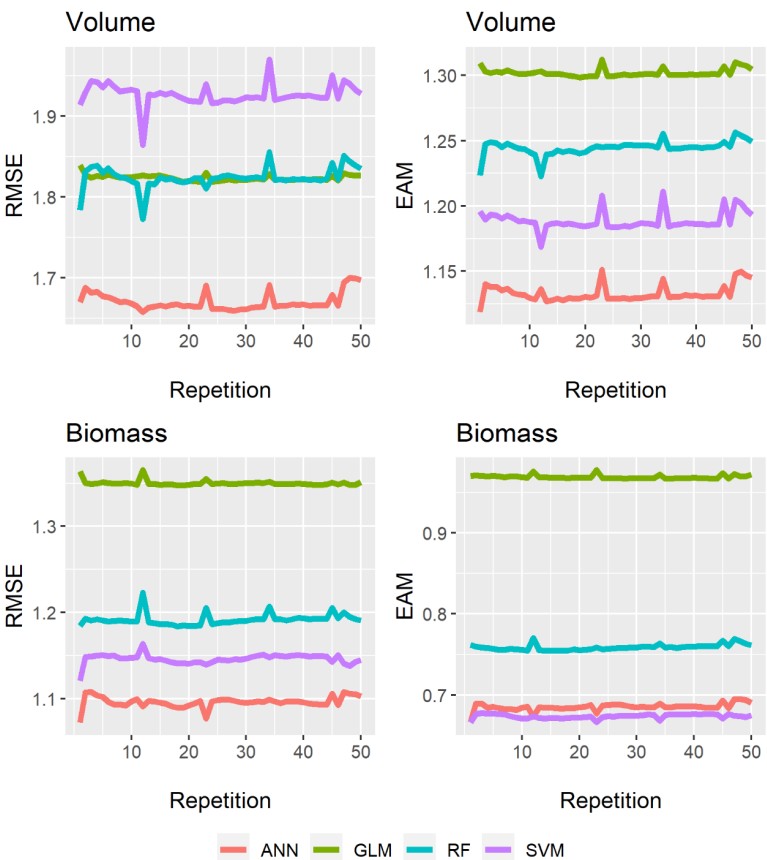

**Figure 4.** Root Mean Square Error (RMSE) of the machine learning models SVM, ANN, RF and GLM

in the modeling of the volume and biomass of commercial trees in the Southwestern Amazon in the municipality of Porto Acre, Acre, Brazil.

The Friedman test with the means of cross-validation RMSE showed that the predictions of the volume and biomass of commercial trees in the Amazon differed between the techniques ($p < 0.05$). Thus, the hypothesis that at least one average of one of the techniques differs from the others was accepted. The Nemenyi test indicated that the difference between the GLM model and the other techniques was greater than the calculated critical difference (CD) when estimating the volume of the trees. The calculated critical difference (CD) of the ANN was greater than the other machine learning techniques evaluated when estimating the biomass (Figure 5).

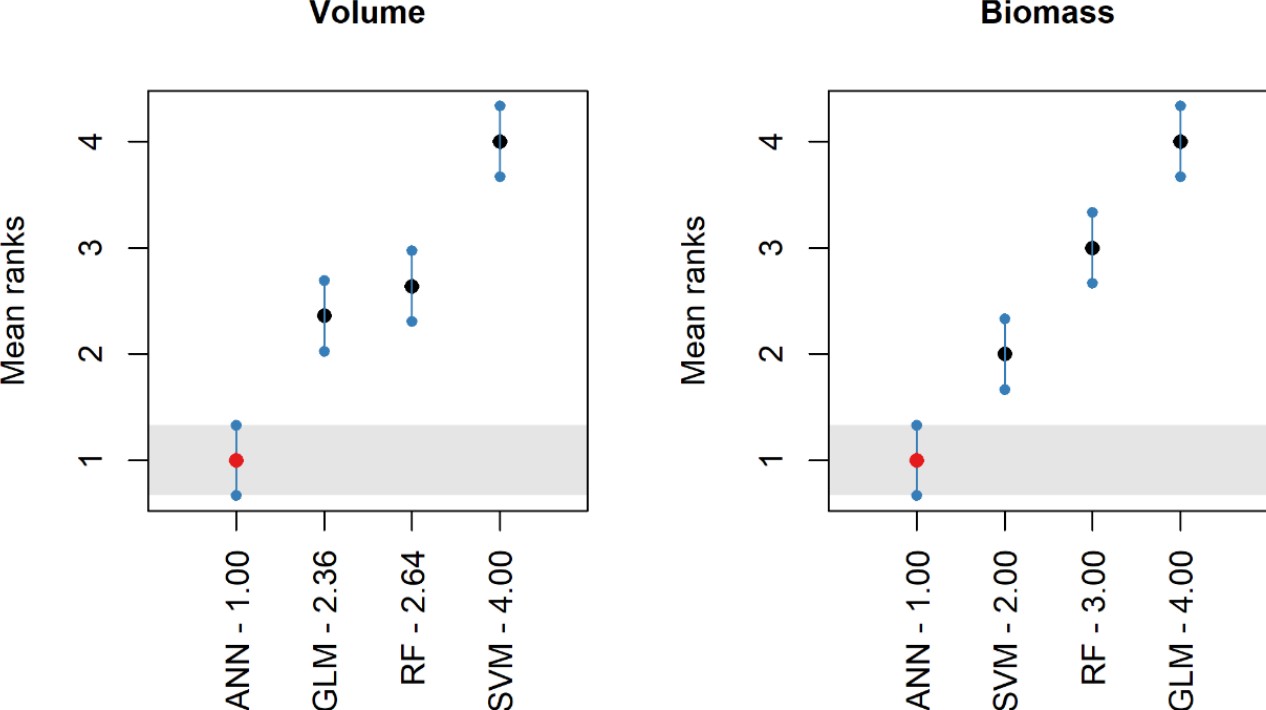

**Figure 5.** Nemenyi test in the cross-validation of the estimates of volume and biomass of commercial trees in the Southwestern Amazon in the municipality of Porto Acre, Acre, Brazil.

## 4. Discussion

### 4.1. Selection of Variables

The Boruta variable selection method is the preferred algorithm among the variable selection methods because it has a high computational efficiency for working on data sets with many predictors. The use of the Boruta algorithm for variable selection offers several advantages including model independence, robustness against multicollinearity, the ability to handle variables of different types, identification of irrelevant variables and statistical evaluation of variable importance [54]. One of the main advantages of Boruta is that it can handle noisy data and correlations between variables, which is particularly useful in real-world problems.

In this study, the method indicated that the diameter and height are fundamental to explaining the allometric attributes of the trees. The stem diameter is a good predictor [7,55,56] and this is an important advantage for practical use. However, the integration of tree height significantly reduces uncertainties [5,10,57]. The diameter of a tree is a critical variable, as it has a strong relationship with tree volume. The height of a tree is also important, as it can affect the distribution of the crown, which impacts the volume of the tree.

Besides that, the inclusion of the wood basic density variable is an important predictor for biomass. This variable, with the diameter of the trunk, the height of the tree and the type of forest (dry, moist or wet), are the most important predictors of the biomass [8]. The inclusion and combination of both provide better quality in adjustment and estimates [8,10].

Our analysis also showed that the bioclimatic and topographic variables were not significant for estimating volume and biomass in commercial trees in the Amazon. This is due to the low variability of this information in the area, in view of the uniform distribution of these characteristics in the study region. However, considering the role of climate in predicting forest attributes can provide more accurate estimates [5,58], since diameter-height relationships in trees depend on a series of physiological and environmental factors [59,60]. The maximum and minimum temperature, precipitation seasonality and degree of solar radiation have strong correlations with biomass [61,62].

### 4.2. Model Performance

In general, the machine learning models accurately estimated the production attributes of the evaluated commercial trees in the Southwestern Amazon. A major advantage of using machine learning methods over traditional models is their applicability to any number of variables [63]. This method is a very valuable procedure for working with data sets in large-scale databases [64,65] because it can manipulate continuous, categorical and binary data [66] and is able to adapt to complex and non-linear relationships between variables, in addition to dealing with interaction effects between them [67]. These models can encompass several types of information and it is possible to work with a single model for different situations.

The Friedman test confirms ($p < 0.05$) that the ANN model gave the best estimates of the volume and biomass of commercial trees in the Southwestern Amazon. The RMSE and MAE corroborate this statement, showing small differences in training and validation. ANN is considered an important non-parametric algorithm for estimating the biophysical parameters of the forest [21]. Neural networks can implicitly detect any complex nonlinear relationships between independent and dependent variables [68]. In contrast to conventional parametric approaches, ANN does not require any assumptions about the statistical distribution of the data.

RF presented intermediate results for estimating the volume and biomass variables. This method produces the most accurate and stable predictions [69]. This algorithm has been widely applied in ecological studies, as it can work with complex data analyses [67]. In addition, RF has been considered one of the best methods of classification and regression due to its high precision for estimation results, high calculation speed, robustness and the ability to predict important variables [70]. Decision tree-based algorithms are easy to apply since fewer parameters need to be estimated. Therefore, they have a high degree of automation [71].

SVM presented less precision in the volume estimates compared to the RMSE and MAE values. SVMs have the inconvenience of a delicate and computationally expensive hyperparameter adjustment. In addition, results for SVM compared to other methods showed only average accuracy. ANNs and RF generally produce better results than SVMs for regression tasks. The simple statistical procedures and the set methods were very competitive for classification [72].

GLM showed less precision in the biomass estimates. This may be related to the link function used. GLMs have the characteristic of being able to choose the residual distribution family, which is important in the case of non-parametric models such as those that follow a Poisson distribution and negative binomial errors [73]. The choice of one function over another may explain the low performance.

As forest volume and biomass are important for forest management, global change monitoring and the modeling of forest productivity, there is a need for reliable methods of assessing and monitoring forest production [21]. The results presented here suggest a

new alternative to predict these forest attributes, providing valuable information for forest managers and for the formation of public policies.

Machine learning models offer a powerful solution to the challenges posed by high variability and complexity in the Amazon Forest ecosystem, enabling accurate estimations of tree volume and biomass. This is crucial for effective forest management and decision-making regarding resource utilization and conservation efforts in the face of deforestation and land-use change.

A limitation of our study is the limited spatial variability of the climatic and topographic data used. However, future research utilizing larger and more comprehensive datasets holds the potential to further enhance the estimation of biomass and volume by incorporating a broader range of spatially diverse environmental variables such as soil characteristics and vegetation indices, leading to improved accuracy and precision. Machine learning's advantage lies in its ability to handle complex and high-dimensional variables that are difficult to incorporate into traditional models. By leveraging machine learning algorithms, researchers can easily include a wide range of variables with intricate relationships, resulting in more accurate and comprehensive estimations.

Overall, while machine learning offers promising opportunities for predicting tree volume and biomass in the Amazon, there are several challenges that need to be addressed to ensure that these models are accurate and reliable. These challenges include selecting appropriate features, choosing the best algorithm and optimizing the model parameters to achieve the best performance. Another challenge is the limited availability and quality of training data, particularly in remote areas of the Amazon. Accurate predictions require large and representative datasets that capture the diversity of tree species and environmental conditions in the region.

## 5. Conclusions

The tested machine learning methods (SVM, ANN, RF and GLM) are useful and efficient tools for estimating the volume and biomass of commercial trees in the Southwestern Amazon. This study represents a new approach to estimating these attributes linked to forest production. ANN is the most suitable for estimating the volume and biomass of commercial trees.

Further research is needed to improve the accuracy of the model and to test its applicability to other tree species and forest types. To achieve this, new techniques, additional variables and larger and more diverse datasets should be explored, along with efforts to minimize biases and uncertainties in the ground-truth measurements. These efforts may lead to more accurate and reliable estimates of tree volume and biomass, which are crucial for sustainable forest management and conservation.

**Author Contributions:** Data collection, writing, methodology, formal analysis, figures—S.J.S.S.d.R. and F.M.B.R.; data collection, review, editing, supervision—C.M.M.E.T., L.A.G.J., S.C.R., P.H.V., B.L.S.S. and V.T.M.d.M.J.; review, editing, supervision—L.P.R., I.B.C., I.d.S.T.J., Á.B.T.V. and M.P.M.X.R. All authors have read and agreed to the published version of the manuscript.

**Funding:** Coordenação de Aperfeiçoamento de Pessoal de Nível Superior—Brazil (CAPES) (88882.437301/2019-01; 88881.311787/2018-01 and 88887.319055/2019-00), Conselho Nacional de Desenvolvimento Científico e Tecnológico—Brazil (CNPq) and Fundação de Amparo à Pesquisa do Estado de Minas Gerais—Brazil (FAPEMIG).

**Institutional Review Board Statement:** Not applicable.

**Informed Consent Statement:** Not applicable.

**Data Availability Statement:** The supporting datasets are stored within the institutions of the authors and available on demand.

**Acknowledgments:** This research was supported logistically by Empresa Fox Laminados Ltd.a. We thank the company's owner, Antônio Aparecido Barlati and his team for their help in the planning, harvesting and sawmill stages of the study. We thank the Laboratory of Wood Anatomy of the Federal

University of Acre (headed by Moisés Silveira Lobão) and the Laboratory of Wood Panels and Energy of the Federal University of Viçosa (headed by Angelica de Cássia Oliveira Carneiro). This work was supported by the Conselho Nacional de Desenvolvimento Científico e Tecnológico (CNPqand Coordenação de Aperfeiçoamento de Pessoal de Nível Superior—Brazil (CAPES)—CAPES/PrInt [88881.311787/2018-01 and 88887.319055/2019-00].

**Conflicts of Interest:** The authors declare no conflict of interest. Supporting entities had no role in the design of the study; in the collection, analyses or interpretation of data; in the writing of the manuscript, or in the decision to publish the results.

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
