# Peer review of "Machine Learning: Volume and Biomass Estimates of Commercial Trees in the Amazon Forest"

_sustainability, doi:10.3390/su15129452_

Round 1
Reviewer 1 Report
The title of the article is clear.
The content of the abstract corresponds to the structure of the article.
The study conducted by the authors is interesting and relevant, but the article needs improvement.
1. Add a few more keywords.
2. I suggest the authors expand the literature review on the use of ML methods for biomass estimation. It is necessary to substantiate the possibility of using ML tools to solve similar problems.
3. One of the most important stages in solving regression problems using ML tools is the selection of model input variables. I think the authors did not describe the variable selection process well enough.
4. Is it necessary to justify why exactly such variables were chosen?
5. It is also necessary to explain how these variables affect the output value.
6. How much data was in the set? How were they divided into training, testing and validation sets?
7. Line 153. How did the authors establish that the dependence is linear?
8. Why the authors used two error indicators, RMSE (Eq. 1) and MAE (Eq. 2). Both metrics are calculated based on the same input data.
9. In the explanations of all formulas, it is necessary to replace "=" with "-".
10. The results are poorly described, fig. 2 and 3 were not understood. It is necessary to comment more qualitatively on the obtained results.
11. In the conclusions, it is necessary to describe the obtained results, directions of further research, limitations of this research, etc.
12. Also in the introduction, the authors noted three research questions of the study. It would be logical to answer them.
Author Response
Thank you for your detailed feedback on my manuscript. I have carefully considered your suggestions and have made the necessary revisions to improve the clarity and coherence. I also appreciate your suggestion and have added the necessary information.

Reviewer 2 Report
sustainability-2284403-peer-review-v1
Machine learning: Volume and biomass estimates of commercial trees in
the Amazon forest.
The authors share a study on Volume and biomass estimates of commercial trees in the Amazon forest using machine learning methods. The topic seems to be interesting. However, the paper lacks a solid reasoning and discussion with respect to the presented claims and findings. Further, too many details are unclear and there are too many grammatically incorrect sentences.
Moreover, the paper has been very poorly organized and presented. It is unconvincing to me the innovativeness and quality of the paper is sufficiently significant to considered to be published in this journal. The abstract of the paper reads poor. It should tell broad scope of the subject of the paper, gap between theory and practice and then what the paper does about it, some key results obtained.
The authors implemented the Boruta algorithm for determining volume and biomass of sample trees using dendro-metric, climatic and topographic variables. However, it is vague and unclear what are the idea behind of the proposed algorithm. How do you guarantee this algorithm gives optimal solution for the corresponding problems? What are the time complexities of these algorithms? The authors should provide a valid proof to show that the solution generated by the proposed algorithms is in fact an optimal solution of the corresponding problems.
The authors should compare the solution and the execution time of the proposed methodology with any other popular solution technique.
The description and validation of the data set and experimental processes, which are essential and valuable components for a scientific paper, are missing.
Conclusion section does not provide any valuable information. The authors fail to summarize the findings, highlight the practical and theoretic contributions. Besides, the limitations and future research directions should be better described.
***
Author Response

(The authors gave the same response as above.)

Reviewer 3 Report
The study involves machine learning models and techniques for estimating volume and biomass of forests, and comparing the models to derive the best fit. This manuscript and the methodology adopted has high potential due to the capability of machine learning models to assess the forest biophysical parameters like volume and biomass. This is a good quality research with good understanding of the concepts involved. Only suggestion is the integration of satellite observations in the analysis, though out of the scope of the current study, but the authors can think about it in future.
Author Response

(The authors gave the same response as above.)

Reviewer 4 Report
Title. The title conveys the main message of the paper — the issues addressed and the relationships among the issues. On the other hand, this answers three important questions: What? How? Where?
Abstract. The abstract is concise, provides a clear overview, and includes essential facts for the paper but it’s necessary to include the aim of the research.
Keywords. These aren’t enough for the topic. I suggest including others.
Introduction. In the first and second paragraphs, the authors talk about the importance of the Amazon rainforest. This is the background necessary to see the particular topic of the research in relation to a general area of study. In the third paragraph, the authors give the background information to understand the study, particularly for commercial trees. In the fourth paragraph, it was highlighted the innovative method of statistical analysis using machine learning models. Finally, in the last paragraph, the aim states the specific purpose of the research, and the research questions help to understand better the study.
Material and methods. In this section, the authors describe the correct steps that were followed during conducting their study.
L93 to 94 — At the end of the paragraph, it’s necessary to indicate Table 1 in order to know which species were studied.
L112 — Change km2 to km2
Results. This section was well written and shows all data with good descriptions. The results must say about the objective that motivates the research.
L188 — DAP wasn’t previously defined. It must be DBH.
L196 — This subtitle was pointed out previously. It must be ‘selection of models’.
Discussion. In this section, the authors take a broad look at their findings and examine the work in the larger context of the field.
Conclusion. This section included the major conclusions, which were briefly written.
L328 to 329 — The authors must indicate that bioclimatic and topographic variables aren’t significantly important for predicting volume and biomass in the region studied.
Tables and Figures.
L101 — It must be number 1 (Table 1)
Figure 2 — Change Hc per Ch.
Author Response

(The authors gave the same response as above.)

Round 2
Reviewer 1 Report
I recommend publishing the article in present form
Author Response
Answer: We would like to express our gratitude for taking the time to review our work and provide feedback.
We are thrilled to receive your recommendation to publish the article in its present form. It is truly encouraging to receive such positive feedback from an expert in the field, and we appreciate your time and effort in reviewing our work.
We greatly appreciate your feedback and constructive criticism, which helped us improve the quality of our research.
We are writing to let you know that we have submitted the article for professional editing to ensure that the English language meets the highest standards. We understand the importance of having our article written in clear and accurate English to effectively communicate our research findings to the scientific community. Therefore, we have taken this extra step to ensure the language is as polished as possible.
We would like to express our gratitude for your time and effort in reviewing the article and providing valuable feedback. We believe that your expert comments have significantly improved the quality of our work.
Overall, we believe that the revisions have significantly improved the manuscript, and we hope that the reviewers will find it suitable for publication in your esteemed journal.

Reviewer 2 Report
sustainability-2284403-peer-review-v2
Machine learning: Volume and biomass estimates of commercial trees in
the Amazon forest.
The revised paper has not been improved, and I think the authors' attitude to the manuscript is rather casual. I do not see anything new such as the appropriateness, novelty and general significance. There are also several technical content and quality issues. It is suggested that the editor remove this manuscript from the evaluation process.
However, the following comments may be useful to the authors:
Lines 43-51, in introduction, before starting the mentioned references, there is a need to add 8-9 lines related to the subject of the paper and write in general introduction. After that you should connect them with the references. Lines 43-83, the introduction is barely sufficient since it does not analyse the state of the art, it seems to me just a list of related papers, there is no insight in the introduction. In particular, the paper is not well written, it is hard to understand and the idea behind the proposed approach seems not enough as contribution to the state of the art. The manuscript, in its current form, requires more effort in the actual presentation, which is fundamental in order to properly evaluate the merit of the research. Line 33, author mentions, “The results showed that the…”. Abstract of the paper should be written in present tense. Lines 74-83, the study-organizing paragraph is missing. Author(s) should include a 'study organizing paragraph' in the end of the introduction section. The paragraph helps reader to understand the overall sequence and flow of your paper. There are many language issues, for example, line 439, in reference [3], the first author name should start with capital letter.
***
Author Response
Answer: Thank you for taking the time to review our manuscript. We appreciate your honest feedback and understood your concerns regarding the quality and presentation of our work.
We apologize for any inconvenience caused. We took your comments seriously and tried to improve the quality of our research. However, we understood that the revised paper did not meet your expectations and did not add anything novel or significant to the existing literature.
We appreciated the comments you provided, which we used to improve the quality of our manuscript further. We worked on addressing the technical content and quality issues mentioned by you. Specifically, we added a study-organizing paragraph at the end of the introduction section to help readers understand the overall sequence and flow of the paper.
Regarding the language issues, we carefully proofread the manuscript and ensured that all the references were formatted correctly.
We have submitted the article for professional editing to ensure that the English language meets the highest standards. We understand the importance of having our article written in clear and accurate English to effectively communicate our research findings to the scientific community. Therefore, we have taken this extra step to ensure the language is as polished as possible.
We also modified the introduction section by adding more lines related to the subject of the paper, as suggested by you.
Thank you once again for providing your feedback, which we valued greatly. We understand that our manuscript may not have been suitable for publication in its previous form, and we took appropriate measures to address the issues raised by you.

Round 3
Reviewer 2 Report
The paper has not been improved up to mark. In addition, the author’s responses are not satisfactory.
Author Response
Dear Editor,
Thank you for the opportunity to revise our manuscript titled " Machine learning: Volume and biomass estimates of commercial trees in the Amazon Forest". We appreciate the valuable feedback from the reviewers, and we have carefully considered their comments and suggestions in revising the manuscript.
We have made the following revisions to address the reviewers' comments:
Reviewer 2:
Comment 1: The paper has not been improved up to mark. In addition, the author’s responses are not satisfactory.
Answer:
Thank you for your email. I'm sorry to hear that you feel the paper has not been improved up to the mark and that the author's responses have not been satisfactory.
I want to inform you that since the paper was initially submitted, we have made significant improvements to it. We have taken into account the feedback from reviewers and have revised the paper accordingly. However, we are aware that there is always room for improvement and we appreciate your feedback on how we can further improve the paper.
We have submitted the article for professional editing to ensure that the English language meets the highest standards. We understand the importance of having our article written in clear and accurate English to effectively communicate our research findings to the scientific community. Therefore, we have taken this extra step to ensure the language is as polished as possible.
We are attaching the English correction certificate for your reference.
We would like to express our gratitude for your time and effort in reviewing the article and providing valuable feedback. We believe that your expert comments have significantly improved the quality of our work.
Overall, we believe that the revisions have significantly improved the manuscript, and we hope that the reviewers will find it suitable for publication in your esteemed journal.
Thank you again for your time and consideration. We look forward to hearing from you.
Sincerely,
Samuel José.
